# Identifying a common backbone of interactions underlying food webs from different ecosystems

Bernat Bramon Mora[1], Dominique Gravel[2,3], Luis J. Gilarranz[4], Timothée Poisot [2,5] & Daniel B. Stouffer [1]

Although the structure of empirical food webs can differ between ecosystems, there is growing evidence of multiple ways in which they also exhibit common topological properties. To reconcile these contrasting observations, we postulate the existence of a backbone of interactions underlying all ecological networks—a common substructure within every network comprised of species playing similar ecological roles—and a periphery of species whose idiosyncrasies help explain the differences between networks. To test this conjecture, we introduce a new approach to investigate the structural similarity of 411 food webs from multiple environments and biomes. We first find significant differences in the way species in different ecosystems interact with each other. Despite these differences, we then show that there is compelling evidence of a common backbone of interactions underpinning all food webs. We expect that identifying a backbone of interactions will shed light on the rules driving assembly of different ecological communities.

[1] Centre for Integrative Ecology, School of Biological Sciences, University of Canterbury, Christchurch 8041, New Zealand. [2] Québec Centre for Biodiversity Sciences, McGill University, Montréal H3A 0G4, Canada. [3] Canada Research Chair on Integrative Ecology, Départment de Biologie, Université de Sherbrooke, Sherbrooke J1K 2R1, Canada. [4] Department of Evolutionary Biology and Environmental Studies, University of Zurich, 8006 Zurich, Switzerland. [5] Département de Sciences Biologiques, Université de Montréal, Montréal H3T 1J4, Canada. Correspondence and requests for materials should be addressed to D.B.S. (email: daniel.stouffer@canterbury.ac.nz)

The structure of ecological networks—the way interactions are distributed among consumers and resources—has been shown to vary in space and time[1,2]. Known drivers of this variation are that species composition is affected by environmental conditions, dispersal limitations, and historical contingencies[3–5]. Ecological interactions also vary over time and from one location to another in accordance with local changes in species abundances and traits[2], as well as due to other intrinsic processes producing ongoing extinctions in the absence of perturbations[6]. The nature of environmental variability in different habitats might also shape ecological networks in different ways. For instance, communities experiencing high seasonality, such as stream and lake food webs, present a strong latitudinal gradient in the number of prey and predators per species[7]. Moreover, the effects of disturbances like invasive species and habitat fragmentation can introduce additional variability that can also lead to changes in network structure[8–10]. Differences in the sampling methods can also lead to changes in the data collected, with some techniques making it hard to observe weak links in particular[11].

Despite this observed variability, many types of ecological networks also showcase a variety of common structural properties across environments[7,12,13]. For food webs, examples include relatively short food chains[14] and a roughly constant fraction of top, intermediate, and basal species[15]. The observation of these common structural properties might suggest the existence of general rules driving or constraining the assembly of all ecological communities[16–20]. Some such rules are thought to be the result of energetic or metabolic constraints in the way individual organisms process energy and materials, which could translate into some of the scaling relations observed across ecosystems[21,22]. Aspects of network structure have also been linked to ecosystems' robustness to species extinctions[23,24], persistence[25–27], and dynamical stability[28,29], which has led to some arguing that stability and feasibility are additional constraints shaping these ecological communities[30].

Notably, the aforementioned structural variability and commonality observed across environments need not be incompatible, though they are often treated as such[31]. Indeed, one heretofore unexplored idea that could reconcile these two perspectives is the existence of a common "backbone of interactions" underlying all ecological communities. Conceptually, this backbone would constitute a set of connected species within every network that play similar ecological roles and that also interact with each other in a similar manner. Extrinsic and intrinsic differences, like environmental variability or variation between local species pools, would then introduce idiosyncrasies in realized community assembly and add noise to and around the backbone.

While a backbone of interactions shared across disparate food webs might be a compelling idea, current methods for comparing network structure across communities lack the power to identify such a level of organization. In particular, existing methods are generally based around the comparison of a library of different descriptors of network structure[23,32–34]; however, these descriptors are summary statistics at the network level and mostly overlook the actual way ecological interactions are distributed within a network. Alternatively, one potential way to identify a backbone is by directly aligning networks in such a way as to pair up species from the different communities that play similar ecological roles (Fig. 1a). Doing this network alignment across a large enough dataset, the backbone of interactions could emerge as a substructure that is consistently aligned across environments.

Recent advances in network science have provided multiple methods for aligning complex networks[35–39]. Most of these methods, however, focus on aligning undirected networks, making them ill-suited for ecological networks like predator–prey food webs in which the direction of interactions is particularly

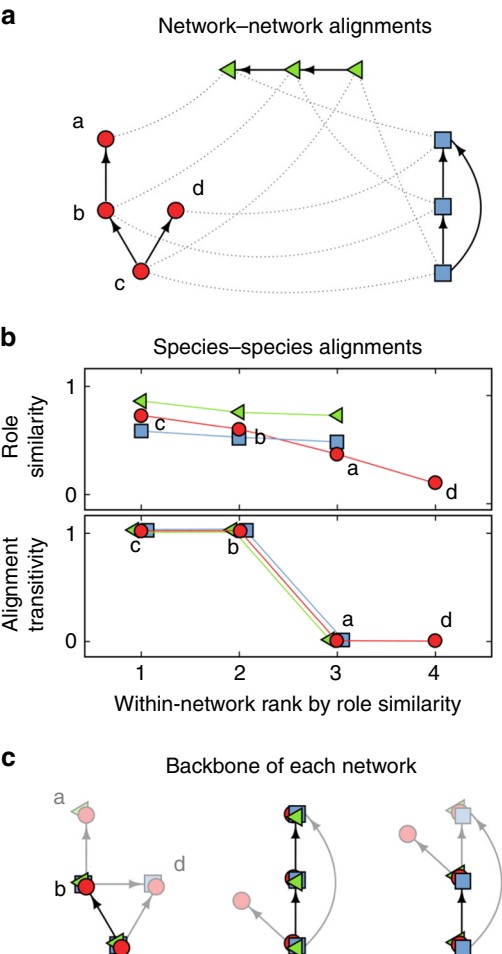

**Fig. 1** Network alignment and identifying a backbone of interactions. **a** An example of the optimal alignments between three simple networks. The red circles, green triangles, and blue squares represent the species in each network, and the arrows indicate the direction of energy flow between those species. The dotted lines characterize the pairings of species in the three alignments between networks. **b** Given the alignments in **a**, we rank species according to the average role similarity that they present across their pairings. The top panel shows the actual average role similarity, and the bottom presents the alignment transitivity of those same species. The best-aligned species from the red network is species c, whereas the worst-aligned is species d. The species in the blue and green networks to which species c is paired are also paired, which implies that the alignment transitivity of c is 1. In contrast, the alignment transitivity for species d is 0 because there are no paired species in the blue and green networks to which d is paired. **c** Given the alignments in **a**, we can also identify the backbone of interactions for each network. Here, the dark links are those that present the maximum overlap across network alignments and therefore characterize the backbone of interactions. The lighter links represent the periphery of such backbone

relevant[40]. In this study, we develop a new alignment technique specifically designed for directed networks, and we then use it to test whether or not there is a backbone of interactions across food webs. For this test, we align a collection of over 400 food webs that were compiled from multiple ecosystems—including different types of freshwater, marine, and terrestrial ecosystems. For every pair of food webs, our method matches their constituent species based on their role similarity, which measures how similarly any two species are embedded within their respective communities. In particular, our method provides us with two key

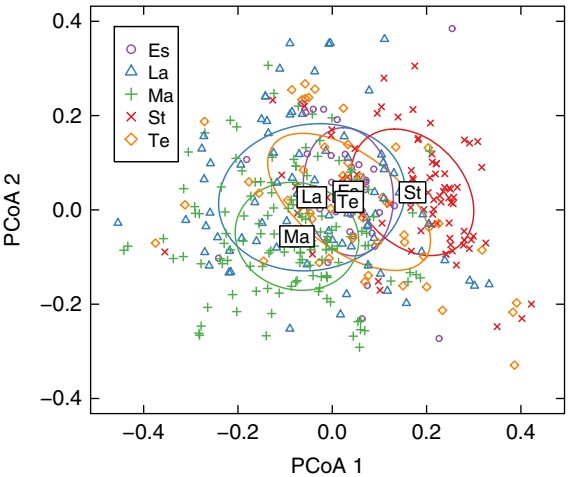

**Fig. 2** Principal coordinate analysis of the dissimilarity matrix $\hat{E}$ containing the normalized pairwise distances between all food webs. Each different color represents the group of networks from estuaries (Es), lakes (La), marine (Ma), streams (St), and terrestrial (Te) ecosystems. The ellipses characterize the 1 standard deviation ellipses about the group medians

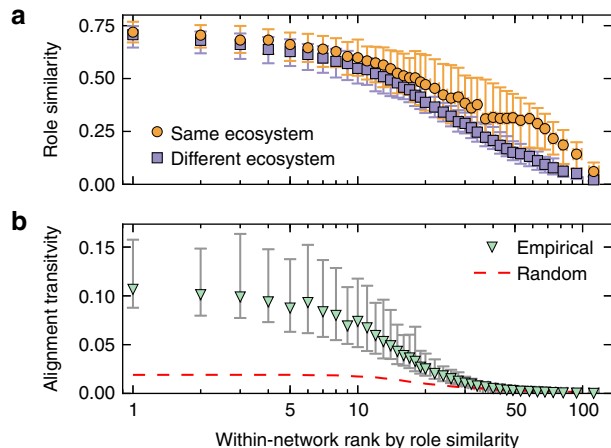

**Fig. 3** Ranking of species from our dataset of 411 food webs based on the average similarity between their role and the roles of the species to which they are paired across all 84,255 alignments. The top panel **a** shows the observed role similarity for all species when compared to food webs from either the same (circles) or different (squares) ecosystem types. The bottom panel **b** shows the alignment transitivity observed for all species across all food webs. The red dotted line represents the expected alignment transitivity for shuffled alignments, where the number of pairings per alignment was maintained. In both panels, every point indicates the median across at least 250 species with the exception of the last point which is the median across 30 species, and the error bars characterize the interquartile range

pieces of information. The first is a metric describing the "quality" of the alignment between food webs, which represents an overall measure of how similar two networks are to each other. The second consists of a list of the corresponding species–species pairings between those food webs, specifying the actual mapping of the alignment between them.

Here, we use the alignment quality as a metric with which to test for structural differences across ecosystems. In particular, we find that food webs from different ecosystems present significantly different network structures. We then leverage the lists of species–species pairings to identify subsets of species within every food web that align better than the rest, since these species could well constitute a backbone (Fig. 1b). Next, we test whether or not these subsets of species are actually linked together, and we observe that they do indeed form a connected backbone of interactions (Fig. 1b). To determine what these connected backbones actually look like, we finally explore the overlap of the backbones between all aligned networks to reveal the hidden structures that underly our dataset (Fig. 1c).

## Results

**Structural differences across ecosystems**. We first analyzed the overall differences across all food webs in order to test whether or not there are significant structural differences across ecosystem types. To do so, we identified optimal alignments between every pair of food webs in our dataset, where each alignment pairs up species with similar interaction patterns in their respective networks (Methods). For each pair of food webs, we started with a random alignment and then used a simulated-annealing algorithm to progressively minimize an alignment cost function that decreases when both paired species and those species' neighbors play similar ecological roles (Methods; Alignment algorithm and Algorithm tests sections of Supplementary Methods; Supplementary Figs. 10–13).

From these pairwise alignments between all food webs in our dataset, we constructed a food-web dissimilarity matrix $\hat{E}$, where every element $\hat{e}_{ij}$ represents the "alignment quality" between any two webs $i$ and $j$ (Methods). Using this matrix, we tested whether or not the alignments between food webs from the same type of ecosystem tend to be better than the ones between food webs

from different ecosystem types. We found that there are indeed significant differences in the quality of the alignments between the different ecosystems (PERMANOVA; $F_{4,411} = 22.81$, $p < 0.01$; Methods). In general, this is true regardless of the choice of alignment quality metric or constraining our dataset to avoid comparing food webs with very different sizes (Fig. 2 and Supplementary Table 1; Structural differences across ecosystems section of Supplementary Note 1). We repeated the tests separately for every pair of ecosystem types in our dataset, finding that the majority of pairwise comparisons reinforced the idea of structural divergence between ecosystems (Supplementary Table 2 and Supplementary Fig. 1; Pairwise comparisons between ecosystems section of Supplementary Note 1). Based on those comparisons, the structure of freshwater stream food webs seems to be the most different when compared to all other ecosystem types.

**Identifying backbones of interactions across food webs**. We next studied the way that individual species from different food webs were matched to each other by collectively analyzing every species–species pairing across network alignments. For every network, we ranked its species based on their average role similarity; that is, based on the average similarity between their role and the role of the species to which they were matched (Methods). Within these rankings, species that match very well— because they have very similar structural roles—will be ranked first, whereas those that present a lower role similarity in their matchings will be ranked last (Fig. 1b). We observed that species' average role similarity can vary considerably (Fig. 3), with some species tending to align substantially better than others. Importantly, this result is independent from the ecosystem type of the food webs. That is, a ranking made based solely on the alignments of food webs within one ecosystem type is generally very similar to a ranking based solely on alignments across different

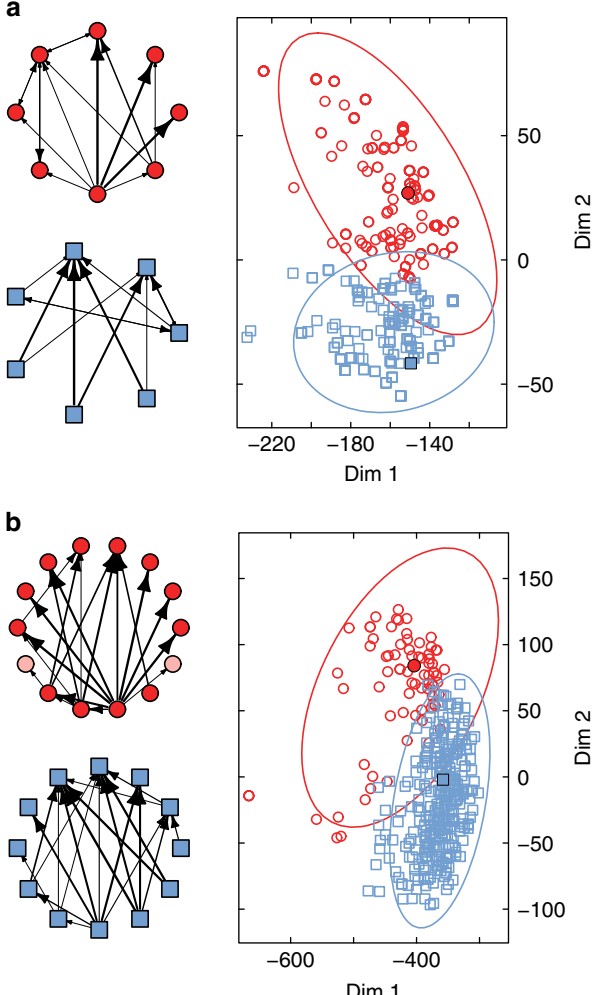

**Fig. 4** Visualization of the backbones of interactions found across all food webs. **a** Analysis for the 6-link backbones of interactions. On the right, we show a representation of the clustering analysis for the dissimilarity matrix $E_6$, where every point represents the backbone from a different network. The red and blue network structures depicted on the left characterize the distinct backbones identified within each of the two clusters. They are found by selecting the medoids of the clusters (indicated by the black circle and square) and overlapping them with all the within-cluster backbones, following the example shown in Fig. 1c. In these red and blue structures, the weight of the links is proportional to the likelihood $l$ of finding them in the backbones. Note that links that were not significantly represented in the backbones ($l < 0.01$) are not shown. **b** We show the same analysis but for the 15-link backbones of interactions. The light-red nodes in the top structure indicate nodes that significantly appears in the backbones but not in the medoid

ecosystems (Fig. 3). Though we previously observed significant differences between ecosystems based on their overall network alignments, the similarity of these species-level pairings implies that the best-aligned species from a given food web will, in general, be the same for any of that web's alignments. Moreover, these species do not exclusively come from a specific trophic level, despite the fact that some trophic levels are vastly over-represented in our data relative to others (Supplementary Fig. 2). They do, however, tend to be those with the greatest total number of interactions (Supplementary Fig. 2).

Our observation that every network has a set of species that align much better than the rest could be indicative of the

existence of a backbone of interactions underlying all these communities. However, this observation is still not a sufficient condition for the backbone to exist. Instead, we identified two necessary conditions for the presence of a backbone of interactions: (i) the best-aligned species from all networks should tend to be paired to each other; and (ii) they should also form a connected component in their own network. To test the first condition, we studied the transitivity of species' alignments, which is a measure of how coherent a species' pairings are across alignments (Fig. 1b; Methods). We observed that the best-aligned species show a significantly higher alignment transitivity than would be expected at random (Fig. 3). This implies that the best-aligned species for the different food webs are in fact paired with each other more often than expected by chance, satisfying condition (i). Next, we indirectly tested the second condition by studying the path likelihood between species, which is a measure of how connected a set of species is within a network (Methods). For every network, we compared the subweb formed by the set of best-aligned species to structures formed by equally sized random subsets of species. We found that the best-aligned species tend to present a high path probability (Supplementary Fig. 3; Connectance and path likelihood section of Supplementary Note 1), which implies that those species are also more connected and likely to form a connected component than expected by chance, satisfying condition (ii).

In satisfying these two conditions, the evidence reveals that there likely is an underlying backbone of interactions across all the food webs in our study. However, these tests do not provide information regarding the shape of such a backbone. To visualize the backbone of interactions, we lastly calculated the link overlap of every network given its full set of optimal alignments (Fig. 1c). Here, the weight of a link between two species is given by the number of times that link is also shared by those species' pairings across all webs. This allows us to identify sets of links that are consistently aligned across networks—much like we previously identified best-aligned species—and to reveal what the backbone of interactions looks like. For a given size $k$, we identified every network's backbone of interactions made up of the $k$ most-overlapped links. Here, we explored backbones in the range $6 \leq k \leq 31$, where 6 corresponds to the network with fewest links in our database and the upper bound 31 ensures that we maintained 75% of the networks in the analyses that follow. This analysis of the backbones' overall structure revealed the most common patterns of interaction forming the backbones (Supplementary Fig. 4). Notably, when examining the 118 food webs for which we had interaction strength data[41], we also observed that the backbones tend to be made up of the strongest links of the community (Supplementary Fig. 5).

To compare the backbones found across food webs, we also aligned them—using the same method as for the full food webs—and generated the corresponding dissimilarity matrix $E_k$ for every backbone size $k$, where $e_{ij|k}$ is the optimal alignment cost between the $k$-link backbones from any network $i$ and $j$ (Eq. (3)). Using clustering techniques, we then analyzed these dissimilarity matrices and identified the number of distinct qualitative structures necessary to explain the obtained backbones of interactions (Fig. 4; Methods). Regardless of the size $k$ of the backbones, we found that we could consistently identify two clusters that characterize the observed backbones (Fig. 4). To find the representative structure for each of these clusters, we identified their medoids and generated the respective overlapping structures characterizing each cluster (Fig. 4 and Supplementary Fig. 6). These structures were consistent with the results found using an alternative measure of network similarity, which does not require aligning the backbones (Links removals: alternative measure for identifying the backbones of interactions section of

Supplementary Note 1; Supplementary Fig. 7). Noticeably, we found the differences between the backbones to be less evident when $k > 15$, which could represent a size or detectability limit for the identification of the backbones in our dataset (Many-link backbones of interactions section of Supplementary Note 1; Supplementary Fig. 8). Finally, we followed the same approach to identify the representative structures for each of the five different ecosystem types. Despite showing some expectable variability, the backbones found independently for every ecosystem type largely agree with the backbones found for the entire dataset (Backbones of interactions for each ecosystem section of Supplementary Note 1; Supplementary Fig. 9).

## Discussion

In this paper, we have developed a new approach to align ecological networks as an attempt to shed light on the way those communities resemble each other while avoiding the loss of information associated with comparing derived measures of network structure. Although it has previously been argued that food webs from different environments share a common set of macroscopic properties[42,43], there is also strong evidence suggesting that food-web structure may differ between ecosystems in characteristic ways[44,45]. One way to reconcile these two findings would be if ecological networks presented a backbone of interactions that was shared across environments. That is, ecological networks could all tend to include a set of ecologically equivalent species that almost always interact in a similar fashion while also showing significant differences in the way the remaining species are attached to the periphery.

To test for the existence of this backbone of interactions, we first focused on detecting actual differences on the alignments between networks from different ecosystems. We observed consistent differences in the structure of food webs across ecosystem types. These differences were particularly strong for freshwater stream food webs, which could well indicate different stability mechanisms associated with high seasonal variability[46]. It is worth noting, however, that we found measurable structural dissimilarities between almost every pair of ecosystem types. Although some of these dissimilarities may potentially be explained away by differences in the sampling methods used to collect the empirical data in different environments[47], the strong consistency of our results for such a diverse dataset suggests a fundamental heterogeneity across ecosystems that has rarely been identified previously[34,45].

Despite finding consistent differences across ecosystem types, we found that within nearly every network comparison there is a set of species in both food webs that present a better alignment than the rest. Those species are also consistently paired across food webs and far more likely to be connected to each other than would be expected at random. These three results combined hint at the idea that there is indeed a backbone of interactions underpinning all food webs. When examining what this backbone actually looks like, we identified the two most-widespread candidates across all networks. Broadly speaking, the two backbones could be described as follows: a structure with high centrality, where few species in the center that are consumed by many satellite species, and a "bipartite" structure, where half of the species are consumers of the other half. Despite the observed differences, species forming each backbone do not seem to belong to distinct trophic levels (Trophic level of the backbones section of Supplementary Note 1). In addition, both backbones were mainly made up of a combination of exploitative competition, generalist predation, and simple three-species food chains[48]. As the size of the backbones increases, we also observed an increase in the number of three-species omnivory loops. We advise

caution, however, when focusing on the topology found for the backbones. Although their existence is crucial to understanding food-web structure, it does not imply that links not found in the backbone are unimportant. Instead, it is best to think that those links are just distributed differently within the networks.

That being said, there are three commonly studied aspects of food-web structure that could be viewed in a different light given our observations of a consistent backbone. First, even though a backbone could appear to be in contrast with the stabilizing effect associated with compartmentalized food webs[26], most of the networks used in this study presented a modular structure (Compartmentalized structure of food webs section of Supplementary Note 1). This suggests that the observed backbones could exist within modules, which could explain some of the noise present in our results. Second, the prevalence of omnivory and its role in the stability of food webs has led to equivocal results. While some work has linked the existence of omnivory to lower stability[49,50], there is strong evidence that suggests a positive relationship when trophic interactions are weak[51,52]. Following this, it is noteworthy that, despite the fact that most of the networks contain three-species omnivory loops, we rarely found this type of interactions within backbones. Regardless of the effect of omnivory interactions on the stability of food webs, this suggests that it is embedded differently across networks. Finally, when considering the networks for which we had interaction strengths, we found that the backbones generally contain the strongest interactions of the community. This may make sense given other correlates of interaction strength. After all, (i) they could otherwise be overlooked in empirical datasets due to sampling errors, and (ii) they might be unable to persist in ecosystems subject to constant environmental change and frequent disturbances.

Among other potential implications of a backbone, we expect that it could be vital to explain and understand food-web dynamics. Similar to the work presented by Murdoch et al.[53], in which they show that the dynamics of generalist consumers can be approximated using one-species models, the backbone of interactions could also be an internal motor that is driving the dynamics of complex ecological communities. Under this perspective, a backbone of interactions could likewise arise as a potential management tool, whereby the dynamics of entire networks could be optimally regulated by focusing on the species forming the backbone[54]. While, it has been shown that the structure of networks might not necessarily influence their functioning[6], the backbone could be a driver that ensures at least minimal functioning by staying intact during ongoing species turnover[55]. Along similar lines, these structures could also arise as useful toy models for the study of how ecosystems react to scenarios of current global change[56]. Further inspection of the species attached to the periphery of the backbones, on the other hand, could potentially provide insights into the mechanisms by which food webs from different environments are shaped under different perturbations[57].

This link between structure and dynamics is especially important because measuring and comparing the topology of ecological networks is much easier than elucidating their dynamics, both empirically and synthetically. Although characterizing the properties of ecological networks and identifying their overall differences across environments have proven to be useful to answer key questions in ecology and evolution[58–60], aligning ecological networks provides a new level of understanding of "how" exactly ecological networks resemble and differ from each other. Consequently, network alignment presents itself as a powerful and versatile tool for the study of ecological communities. The identification of species that are critically affected by environmental perturbations[61], for example, could be used as a strategy for selecting other species from different communities

that might be sensitive to similar disturbances. The empirical observation of the dynamics of one ecological network could then be extended to other networks by simply aligning them together, avoiding the use of mathematical models that might oversimplify the dynamics of these ecological systems[62,63].

Finally, we identify two aspects that stand out as key steps moving forward. First, though computationally intensive, it could be worth testing the existence of a backbone in randomized, as opposed to empirical, communities. These test could reveal the conditions under which different backbone structures emerge[64]. While network properties might significantly change following certain reshuffling processes, backbones could be found to instead persist; it would then be the periphery attached to the backbone that is absorbing the effects of the randomizations[65]. Second and perhaps more important, further exploration of the species that make up the backbone of interactions should provide a very interesting perspective. If there are indeed intrinsic properties such as traits or shared evolutionary history that are common across the species in the backbone, this could shed light on fundamental aspects of community assembly[66]. Importantly, this might not only untangle the eco-evolutionary mechanisms explaining the formation of such a backbone but may also allow us to understand the role of the backbone as a driver of species' coexistence and diversification[67].

## Methods

**Empirical data**. We combined the data from multiple previous studies to build a large dataset of networks sampled from different environments and capture as much empirical variation as possible[7,41]. Because they are incompatible with our methodology, we excluded any bipartite networks; we also limited ourselves to communities ranging in size from 5 to 133 species due to computational difficulties and greater degeneracies in larger networks. In total, we used 411 food webs from 34 estuaries, 87 lakes, 148 marine ecosystems, 88 streams, and 54 terrestrial ecosystems.

**Species role similarity**. To measure the roles of different species, we used the definition based on the idea of network motifs[68]. Network motifs represent the distinct $n$-species subnetworks describing all unique patterns of interactions between $n$ species. It has been shown that one can characterize the role of any given species $a$ based on the number of times $c_{ai}^n$ that it occupies each distinct position $i$ of the $n$-species network motifs[68] (Alignment algorithm section of Supplementary Methods). This definition allows a convenient way to compare the topological roles of different species. In particular, given any two species $a$ and $b$ with motif-role profiles $\vec{c}_a$ and $\vec{c}_b$, we used the Pearson's correlation coefficient to define a "measure" of similarity between them as:

$$\rho(a,b) = \frac{\text{cov}(\vec{c}_a, \vec{c}_b)}{\sigma_{\vec{c}_a} \sigma_{\vec{c}_b}}, \qquad (1)$$

where $\text{cov}(\vec{c}_a, \vec{c}_b)$ is the covariance between roles and $\sigma_{\vec{c}_a}$ and $\sigma_{\vec{c}_b}$ are the standard deviations of $\vec{c}_a$ and $\vec{c}_b$, respectively. This measure of similarity is equal to 1 if $a$ and $b$ play equivalent roles, 0 when there is no correlation between them, and $-1$ if they play opposite roles.

**Identifying optimal alignments**. We define an alignment between two food webs $A$ and $B$ as a set of one-to-one species pairings $\lambda = \{(a,b)\}$. We allow $\lambda$ to contain three different types of elements: a unique pairing $(a,b)$ between two species $a \in A$ and $b \in B$; an element $(a,\emptyset)$ representing an unpaired species $a \in A$; and an element $(\emptyset, b)$ representing an unpaired species $b \in B$. Such unpaired species necessarily arise, for example, if the two networks are of different sizes; in addition, species in $A$ and $B$ need not resemble each other and hence alignments may not be optimal if dissimilar species are paired together.

Following this definition, the cost function associated with any given alignment can be characterized in multiple ways. One possibility would be to simply consider the sum of every individual species–species pairings as in

$$e_{AB}(\lambda) = \sum_{(a,b)\in\lambda} (1 - \rho(a,b)), \qquad (2)$$

where $\rho(a,b)$ is the measure of role similarity defined above, and for which we assign a penalty of $\varepsilon$ for species that remain unpaired (i.e., $\rho(a,\emptyset) = \rho(\emptyset,b) = \varepsilon$). Minimizing this cost function by changing the alignment $\lambda$ should directly result in matching species that play similar roles in their respective communities (Supplementary Fig. 10).

Unfortunately, this strategy for optimizing alignments guarantees that similar species from different food webs are matched based on their own structural roles but does not guarantee that their neighbors are optimally matched, or even that their overall networks are aligned (Alignment algorithm section of Supplementary Methods). To overcome this drawback, we instead use another cost function to pair up species based on the structural-role similarity of their neighbors. That is, two species from different food webs will only be perfectly matched if their neighbors are also matched with equivalent roles (Supplementary Figs. 10, 11). Therefore, the contribution of two paired species to the overall cost function will be the sum across their neighbors' pairings. With this in mind, we define an improved cost function as follows:

$$e_{AB}(\lambda) = \sum_{x\in\lambda} \left( \sum_{(\alpha,\beta)\in\lambda_x} (1 - \rho(\alpha,\beta)) + \xi_x \right), \qquad (3)$$

where, given the pairing $x = (a,b)$ between two species $a \in A$ and $b \in B$, we define the subset $\lambda_x = \lambda_{(a,b)}$ of $\lambda$ as the set of all the one-to-one pairing $(\alpha,\beta)$ containing both a neighbor $\alpha$ of $a$ and a neighbor $\beta$ of $b$. Following this, $\xi_x$ represents the penalty associated with the unpaired neighbors of every pairing $x = (a,b)$, which accounts for both the number of neighbors of $a$ that are not paired with a neighbor of $b$ and the number of neighbors of $b$ that are not paired with a neighbor of $a$ (Alignment algorithm section of Supplementary Methods).

**Alignment quality**. In order for the alignments to be comparable across our dataset, we also need a network-size-independent measure of how good those alignments are. This is because the alignment cost function defined above is useful for optimizing pairwise network alignments but strongly scales with the size of the networks being aligned. Although neutralizing this size effect is non-trivial, there are multiple ways to appropriately reduce the effect of a size difference between networks (Alignment quality measures section of Supplementary Methods). Here, we adopt an approach described as follows. Given the best alignment $\hat{\lambda}$ found between two networks $A$ and $B$, we calculate the normalized dissimilarity $\hat{e}_{AB}(\hat{\lambda})$ between them rewriting Eq. (2) as follows:

$$\hat{e}_{AB}(\hat{\lambda}) = \frac{1}{N} \sum_{(a,b)\in\lambda} (1 - \rho(a,b)), \qquad (4)$$

where we now set the cost associated with an unpaired species to $\rho(a,\emptyset) = \rho(\emptyset,b) = 1$, and $N$ represents the total number of matches between one species from $A$ and one species from $B$. We chose a normalized version of Eq. (2) for alignment quality because it is much simpler than the same for Eq. (3). Other alignment quality measures are also considered in the Alignment quality measures section of Supplementary Methods, Supplementary Fig. 12.

**Quantifying structural differences across ecosystems**. To test for differences across ecosystem types, we analyzed the alignment dissimilarity matrices using a permutational multivariate analysis of variance[69] (PERMANOVA), which expands beyond the traditional analysis of variance methods (ANOVA) and assesses relative differences between and within treatment groups (e.g., ecosystem types) using a permutation-based significance test.

**Alignment transitivity**. The transitivity between alignments characterizes the cliquishness of all species–species alignments. Suppose that we align a set of food webs $\{A, B, \ldots, Z\}$. Given that species $a \in A$ is aligned with species $b \in B$ and $c \in C$, the alignment transitivity of $a$ is the likelihood of $b$ and $c$ also being aligned.

**Path likelihood**. The path likelihood is a useful measure for testing whether or not a set of species of a network form a connected component. Given a network $A$ comprised of $n$ species, the path probability of a subset comprised of $k < n$ species is defined as the probability that at least one undirected path existed between all pairs of $k$ species.

**Number of distinct backbones**. To find the number of different candidate backbones, we used the R package NbClust, which determines the number of clusters that characterize a dissimilarity matrix by means of combining five different indices and eight clustering methods[70]. Given the number of clusters from each index and method, we used the majority rule to identify the actual number of clusters.

**Code availability**. Code to conduct the network alignment described here can be made available upon request.

**Data availability**. Data to conduct the analyses performed here can be obtained following Cirtwill et al.[7] and Jacquet et al.[41], or made available upon request.

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

## Acknowledgements

We thank all those with whose data have made this work possible. For their help and discussions on the project, we thank the members of the Stouffer and Tylianakis Labs. We also thank Jennifer Dunne and the Santa Fe Institute, where we started the discussions that lead to this project. B.B.M. acknowledges the support of a Rutherford Discovery Fellowship (to D.B.S.). D.G. acknowledges the Canada Research Chair and the NSERC Discovery Grant programs. D.B.S. acknowledges the support of a Rutherford Discovery Fellowship, administered by the Royal Society of New Zealand.

## Author contributions

B.B.M. contributed to the design of the work; contributed to the writing of the code to perform the network alignment; performed the research; led the writing; contributed to the revisions; and gave final approval for publication. D.B.S. contributed to the design of the work; contributed to the writing of the code to perform the network alignment; contributed to the revisions; and gave final approval for publication. D.G., L.J.G., and T.P. contributed to the design of the work; contributed to the revisions; and gave final approval for publication.

## Additional information

**Competing interests:** The authors declare no competing interests.

