## [Peer Review File · Nature Communications]

Reviewers' comments:

Reviewer #1 (Remarks to the Author):

The major claim of this paper is the identification of backbones within ecological networks. More precisely, the authors investigate whether food webs from different ecosystems can be seen as a combination of a universal sub-network and individual extensions. The authors compared the structures of more than 300 food webs from various places and identified three types of such backbones. Although I was intrigued by the overall idea of the study and also by the possible applications, I have to admit that I was not fully convinced by the results. The following comments and suggestions hopefully help to clarify potential gaps. I suggest a major revision before a final decision is reached.

major comments / suggestions:

- line 145: After reading the introduction, I was very surprised by the somewhat artificial number of six links within the backbone structures. I would have expected that not only the structure, but also the size of the backbone is something that comes out of this analysis instead of being fed into it. It is completely unclear, whether the results depend on this choice. If we ignore the smallest food web in the data base and assume backbones consisting of 10 or 15 links, would we still observe the same three qualitative types of backbones?

- I was also wondering whether the analysis would lead to similar results when considering weighted instead of binary networks. I know that empirical data with weighted interaction rates is very scarce, but the relative strength of different links within the networks can be crucial to understand its dynamics and stability (see for example Emmerson, Mark, and Jon M. Yearsley. "Weak interactions, omnivory and emergent food-web properties." *Proceedings of the Royal Society of London B: Biological Sciences* 271.1537 (2004): 397-405. or Neutel, Anje-Margriet, Johan AP Heesterbeek, and Peter C. de Ruiter. "Stability in real food webs: weak links in long loops." *Science* 296.5570 (2002): 1120-1123.). Could you please comment on that?

- If the structures that you identified as backbones truly represent universal sub-structures, then I would assume that these substructures contain very strong links and form very stable communities. Otherwise they would often be overlooked in empirical data sets due to sampling errors that neglect weak links or they would not persist within real ecosystems that are subject to constant environmental change and frequent disturbances. So I was wondering whether you have any idea what makes these particular substructures robust and stable? I was surprised to see that you did not discuss this aspect. For example, the first structure is perfectly nested, which is well known to be stabilizing in mutualistic networks, but not in antagonistic networks (see for example Thébault, Elisa, and Colin Fontaine. "Stability of ecological communities and the architecture of mutualistic and trophic networks." *Science* 329.5993 (2010): 853-856.) Why would such a structure form a "stable core" and be omnipresent in food webs? I am not sure, but one way to clarify this point could be to argue via feedback loops. It seems that none of the backbones presented in the main article and in the supplementary material contains a three-level loop (shortest possible positive feedback, see figure 1 in Neutel, Anje-Margriet, and Michael AS Thorne. "Interaction strengths in balanced carbon cycles and the absence of a relation between ecosystem complexity and stability." *Ecology letters* 17.6 (2014): 651-661.)

- Why would we expect a coherent backbone, when it is known that modularity (or compartmentalization) stabilizes food webs? (see for example Stouffer, Daniel B., and Jordi Bascompte. "Compartmentalization increases food-web persistence." *Proceedings of the National Academy of Sciences* 108.9 (2011): 3648-3652.)

- figure 3: How do the 25% of the networks look like that do not contain the three backbones presented here? Is there perhaps a fourth or even a fifth backbone? How many backbones do we need to explain 100% of the empirical data and how many potential backbones with 6 links exist?

- figure 3 / S6: I asked myself whether the three backbones that you present in the main article (1= something nested, 2= something with omnivory and 3= something broom-like) are typical representatives for different ecosystems, but could find no answer. I was looking for something like figure S6 but based on the three backbones presented in figure 3. Instead, figure S6 left the impression that your results are actually not very robust, since every ecosystem has its own three backbones instead of being represented by the ones that are introduced in the figure 3. This left me very puzzled...

- I had an additional idea for a possible application of backbone structures that might be worth mentioning in the discussion. Many researchers currently focus on the question how ecosystems react to disturbances and scenarios of current global change. There are basically three approaches to do this: (1) Simply construct a specific network based on empirical data from a specific study site and analyse its response to changing conditions. Unfortunately, the results are then only relevant for this specific network without revealing any general insights. (2) Use an ensemble of abstract networks (as for example done in Binzer, Amrei, et al. "Interactive effects of warming, eutrophication and size structure: impacts on biodiversity and food-web structure." *Global change biology* 22.1 (2016): 220-227.) in order to develop a more general understanding of the processes at hand. Unfortunately, such a study can become quite costly in terms of computer runtime. (3) Use a toy model that represents a universal, easy-to-handle food web module instead of complex food webs. The most prominent candidate is a food chain or a simple predator-prey system (as for example done in Fussmann, Katarina E., et al. "Ecological stability in response to warming." *Nature Climate Change* 4.3 (2014): 206.), which probably oversimplifies the whole problem. So using a set of network backbones might be a good compromise between being too specific, too complex or too simple. It might provide a good basic understanding of the relevant processes and thus lead to more reliable predictions.

- figure 3: I had the feeling that backbones of only six links are actually too small to be meaningful. So I was thinking about the following test: Could you also check for randomized networks in addition to randomized alignments? More precisely, if you construct "random networks" by reshuffling the interactions within the real networks, would these randomized networks still contain the same backbones? And does the result depend on the algorithm of the reshuffling (e.g. keeping the number of predators or preys or links per species constant)? I think the backbones that you found would greatly gain relevance if you could show that they only occur within real but not within random networks.

minor comments / suggestions:

-In the first paragraph of the introduction, you list a number of prominent examples of processes and disturbances that influence the structure of ecological networks and that potentially explain their differences. To my knowledge, the network structure might also change over time due to intrinsic processes, even in the absence of disturbances and spatial or temporal variability, leading to a broad range of possible network structures within one and the same (undisturbed) environment - See for example Allhoff, Korinna Theresa, et al. "Evolutionary food web model based on body masses gives realistic networks with permanent species turnover." *Scientific reports* 5 (2015).

- Within the same model (Allhoff 2015), it has been shown that the network structure not necessarily influences its functioning. I wonder whether this could be explained in terms of a network backbone

that stays intact during ongoing species turnover and that ensures the overall functioning of the web...? See Allhoff, K. T., and B. Drossel. "Biodiversity and ecosystem functioning in evolving food webs." *Phil. Trans. R. Soc. B* 371.1694 (2016): 20150281.

- First paragraph of the introduction: Another thing that potentially influences the structure of (empirical) food webs is the sampling method. Some methods might ignore weak links, which could be captured using other techniques, leading to completely different networks when using different methods...

- line 132: How do you define "best"? If I understood it correctly, then you use simulated annealing to get the best alignment between two networks and then you simply pick the species that is best aligned. But how do you make sure that your annealing algorithm is not stuck within a local extremum? Maybe another alignment would be even better?

- line 206-209: It is not clear to me how simply aligning antagonistic and mutualistic networks might be of any use. It is well known that their structures differ considerably (see for example Thébault, Elisa, and Colin Fontaine. "Stability of ecological communities and the architecture of mutualistic and trophic networks." *Science* 329.5993 (2010): 853-856.), suggesting that each of the mentioned network types would have a unique set of backbone structures...

- Concerning the material and methods section: I have to say that this section is remarkably nice and clear, although the study design is far from being intuitive for a non-specialist. Nevertheless, I would like to make two suggestions. (1) Unless I missed it, the definition of a species "role" is not given in the main text, which forces the reader to go back and forth between the main document and the supplementary material to understand the methods. Would it be possible to include the definition of the role into the main text? (2) I had to go back and forth between the results and the methods section. If possible, I would move the methods before the results to avoid too much scrolling.

- line 290-292: This is not perfectly clear to me: To get an isomorphic substructure, links have to be removed from both C and D, right? The sentence suggest that links can be removed from C or D...

- figure 2: I was surprised to see a log-scale on the x-axis. If you simply rank all species in the network, then why would you need a log-scale? Apart from that, the figure suggests that your networks contain less than 100 species, although in line 222 you state that you have between 5 and 130 species per network. Could you please clarify this?

- line 4-5 in the supplementary material: The four alignment quality measures have not yet been introduced, so the reference to "above" is confusing.

- line 46 in the supplementary material: As indicated above, I am not convinced that you found "similar candidate backbones" across all ecosystems. Most of them can not be categorized into something nested, something with omnivory and something broom-like...

- figure S1: Can you give an interpretation of the two axis?

- Is figure S3 referenced somewhere in the text?

- figure S5: Do you have an explanation why stream and terrestrial ecosystems are better described by the three backbones than the other ecosystems?

- line 68-69 in the supplementary material: In principle, you could also consider more complicated

modules – why is 3 a good size?

- line 96 in the supplementary material: You previously used a different symbol for NULL.

- line 100 in the supplementary material: "in Supplementary Fig." is missing in the sentence. Same in line 104.

Reviewer #2 (Remarks to the Author):

The present study develops a new alignment technique specifically designed for directed networks such as food webs, which identifies "connected species within every network that play similar ecological roles and that also interact with each other in a similar manner". The authors use this technique to both evaluate differences among 357 food webs across estuaries, lakes, marine, streams and terrestrial ecosystems and also to identify the "backbone of interactions" shared across the food webs. The developed technique is novel and it could greatly influence thinking in the field of ecological networks. Overall, I think this work could become a great contribution to our field.

Major comments

However, the explanation of the technique and results needs to improve. Some key concepts are not well defined (or not defined at all) in the main text and the reader needs to go back and forth several times in between the main text, the Materials and Methods section and the Supporting Information to grasp the key concepts behind the technique and the axes of the main figures. In particular, "role distance" must be CONCEPTUALLY defined in the main text (somewhere in L63-70) and in the figure legends where it appears. It is not enough to send the reader to a mathematical definition in the Supporting Information. I would like to see the same conceptual work that the authors did for "backbone of interactions" in L44-46 applied to "role distance". In fact, I do not think that "distance" is the right term for expressing the intuition behind the concept. I do not see how we could talk about distances between species roles. It seems to me that role similarity would make more sense (making the corresponding adjustments in the formula of Eq. 1 of the Supporting Information and in the figure axes). Besides, the authors should include the mathematical definition of "role distance" (or similarity?) in the Material and Methods section. A second concept that needs clarification is "Link removal". The authors mention it in L149-150 and it appears on the x-axis and legend of Fig 3 with a "(see Material and Methods)". However, the procedure of link removal does not appear in the Materials and Methods section. A third concept that needs to be clarified in the main text is "Transitivity of species' alignments". I suggest providing a conceptual definition in L127 and in the legend of Fig. 1. So the reader can understand the main results and conclusions without going to the Supporting Information.

Other comments

L104-105: clarify

L127-130: Explain why the result "We observed that the best aligned species show a significantly higher alignment transitivity than would be expected at random (Fig. 2)" implies "the best aligned species for the different food webs are in fact paired with each other more often than expected by chance", and why this "satisfies condition (1)". A conceptual definition of alignment transitivity would help to clarify this result.

L133-136: Same problem as in my previous comment on L127-130. Clarify.

L142: Add something like "across all the webs" after "species' pairings"

L149: Remove "zero or"

L160-165: I do not understand how the identified backbones of relatively few species and interactions can explain macroscopic properties such as connectance, links per species, etc. Explain.

Add letter a) and b) to distinguish top and bottom panels in Fig. 2

Supplementary Figure 7b, 4th row: B and b are not Nbrs. of D and d, respectively... shouldn't it be e instead?

Sincerely,
Dr. Fernanda S Valdovinos
Assistant Professor of Theoretical Ecology
Dept. Ecology and Evolutionary Biology
Center for the Study of Complex Systems
University of Michigan

Reply to Reviewer #1

The major claim of this paper is the identification of backbones within ecological networks. More precisely, the authors investigate whether food webs from different ecosystems can be seen as a combination of a universal sub-network and individual extensions. The authors compared the structures of more than 300 food webs from various places and identified three types of such backbones. Although I was intrigued by the overall idea of the study and also by the possible applications, I have to admit that I was not fully convinced by the results. The following comments and suggestions hopefully help to clarify potential gaps. I suggest a major revision before a final decision is reached.

Reviewer #1 was intrigued by the overall idea of the study and also by the possible applications. However, the Reviewer provided several comments on aspects in which the manuscript and analysis need to be improved in order to clarify the potential gaps and gain robustness. In particular, the Reviewer was not convinced with the third part of the analysis where we attempted to identify the backbones of interactions using the measure of link removals. The Reviewer also pointed out a few areas and ideas in the discussion that were missing, as well as provided some useful suggestions on places in which the manuscript could generally be improved.

We thank the Reviewer for these insightful comments. Indeed, some of these comments sparked a major change to the manuscript that we believe makes our results much more robust. Specifically, we changed the way in which we identify and visualize the backbones of interactions, and we added a few additional analyses that help understand the nature of the observed backbones. We summarize each of Reviewer's concerns in turn below, and we hope that our responses (preceded by **R:**) are to the Reviewer's satisfaction.

1) Backbone size.

line 145: After reading the introduction, I was very surprised by the somewhat artificial number of six links within the backbone structures. I would have expected that not only the structure, but also the size of the backbone is something that comes out of this analysis instead of being fed into it. It is completely unclear, whether the results depend on this choice. If we ignore the smallest food web in the data base and assume backbones consisting of 10 or 15 links, would we still observe the same three qualitative types of backbones?

R: We thank the Reviewer for this helpful comment and apologize for the lack of clarity regarding this aspect of the analysis. We acknowledge that the number of links used for the visualization of the backbones is indeed somewhat artificial or arbitrary. For this reason, we decided to add a major change to the manuscript regarding how we visualize

backbones that will hopefully shed greater light on the concerns raised by the Reviewer (L153–176).

In the previous version, one of the reasons for not investigating backbones consisting of 10 or 15 links was because comparing those structures is very non-trivial. In particular, the measure of ‘link removals’ defined in the manuscript, while providing an illustrative way to compare the backbones, is also computationally very expensive, especially when comparing bigger backbones. Therefore, we decided to follow a different strategy and moved the previous analysis that used the measure of link removals to the Supplementary Information as a demonstration of our results’ robustness.

In this new approach, we first extracted all the n -link backbones for every network (from 6 to 29 links). For each size n , we then aligned the corresponding backbones with each other using our alignment algorithm. These alignments provided us with a dissimilarity matrix of backbones for any given number of links. Using different clustering techniques on these dissimilarity matrices, we were finally able to both identify the optimal number of backbones underlying the networks and characterize the representative backbones (L153–176). Importantly, this change in methodology, also proved crucial for the answer to the concern risen by the Reviewer regarding the actual number of backbones, which we address below. We are confident that this new analysis will be to the Reviewer’s satisfaction and we thank them for what has been a very fruitful suggestion.

2) *Weighted networks vs binary networks.*

*I was also wondering whether the analysis would lead to similar results when considering weighted instead of binary networks. I know that empirical data with weighted interaction rates is very scarce, but the relative strength of different links within the networks can be crucial to understand its dynamics and stability (see for example Emerson, Mark, and Jon M. Yearsley. “Weak interactions, omnivory and emergent food-web properties.” *Proceedings of the Royal Society of London B: Biological Sciences* 271.1537 (2004): 397-405. or Neutel, Anje-Margriet, Johan AP Heesterbeek, and Peter C. de Ruiter. “Stability in real food webs: weak links in long loops.” *Science* 296.5570 (2002): 1120-1123.). Could you please comment on that?*

R: We thank the Reviewer for making this comment because it not only speaks to one of the natural next steps of the method presented here, but it also makes a very valid point for understanding the existence of the backbones and their implications. First of all, we would expect that considering weighted networks instead of binary networks would lessen some of the noise observed in our results. The information encoded within weighted networks is much higher than that in their binary counterparts, and this should arguably reduce the number of potential symmetrical alignments existing between networks. To align weighted networks, we would first need to redefine the structural roles

of the species in order to include information regarding the interaction strengths of the links involving them. Although aligning weighted networks is an interesting perspective and we have actually done some preliminary work on an extended method for doing so, this is unfortunately an aspect that goes beyond the scope of this study. That said, we think that the Reviewer's comments on the strength of the links composing the backbones is a very interesting point, and we regret that we did not think more along these lines in our initial submission. Data with weighted interaction strengths is certainly scarce, but we actually had 118 networks in our dataset for which we had this information. We found that links forming the backbones are generally stronger than those attached to their periphery (L158–160). As pointed out by the Reviewer in the comment below, this would be expected because otherwise they would be overlooked in empirical datasets and would rarely persist within systems with constant perturbations. We have included this new result and the consequent discussion in the manuscript (L226–231), and we again thank the Reviewer for such helpful suggestion.

3) *Stability properties of the backbones.*

*If the structures that you identified as backbones truly represent universal sub-structures, then I would assume that these substructures contain very strong links and form very stable communities. Otherwise they would often be overlooked in empirical data sets due to sampling errors that neglect weak links or they would not persist within real ecosystems that are subject to constant environmental change and frequent disturbances. So I was wondering whether you have any idea what makes these particular substructures robust and stable? I was surprised to see that you did not discuss this aspect. For example, the first structure is perfectly nested, which is well known to be stabilizing in mutualistic networks, but not in antagonistic networks (see for example Thébault, Elisa, and Colin Fontaine. "Stability of ecological communities and the architecture of mutualistic and trophic networks." *Science* 329.5993 (2010): 853-856.) Why would such a structure form a "stable core" and be omnipresent in food webs? I am not sure, but one way to clarify this point could be to argue via feedback loops. It seems that none of the backbones presented in the main article and in the supplementary material contains a three-level loop (shortest possible positive feedback, see figure 1 in Neutel, Anje-Margriet, and Michael AS Thorne. "Interaction strengths in balanced carbon cycles and the absence of a relation between ecosystem complexity and stability." *Ecology letters* 17.6 (2014): 651-661.)*

R: We would like to apologize for not going into further detail regarding the actual properties of the backbones that might make them stable. The focus of the manuscript was testing whether or not there is a backbone of interaction underlying food webs, and since we did not perform a detailed analysis on the stability properties of the backbones, we preferred not to speculate too far on this aspect to avoid overreaching. First of all,

we would like to point out that the fact that we don't observe feedback loops on the backbones does not mean that they are not there; it only implies that, if they are present, they are not embedded homogeneously across food webs. Likewise, the backbones of interactions are made of the most overlapped links, which leads us to believe those could be crucial for the stability of the networks. Nevertheless, this does not mean that the rest of links are not important; it only implies that they are distributed differently within the networks. In a similar way to weak links—as shown in Emmerson and Yearsley, *Proc R Soc Lond B* (2004)—, the links attached to the periphery of the backbones could also play a crucial role for understanding the stability of the backbones. In retrospect, we acknowledge that this overall observation is a valuable nuance that we should have addressed in the discussion of the initial submission (see L209–213 of the new version of the manuscript), and we thank the Reviewer for pointing this out. In addition, we agree that the manuscript was missing a deeper look at some of the potential reason for the existence of these backbones. We have now included an analysis of the patterns of interactions that compose the backbones as well as a look at the strength of the links forming the backbone (as discussed in the previous comment). We are confident that this new analyses as well as the additions to the discussion (L219–231) will be to the Reviewer's satisfaction.

4) *Backbones on compartmentalized food webs.*

*Why would we expect a coherent backbone, when it is known that modularity (or compartmentalization) stabilizes food webs? (see for example Stouffer, Daniel B., and Jordi Bascompte. "Compartmentalization increases food-web persistence." *Proceedings of the National Academy of Sciences* 108.9 (2011): 3648-3652.)*

R: We are grateful to the Reviewer for making this point since we agree that the concept of a backbone could seem to be in contrast with the compartmentalized structure observed across food webs (a topic a number of us have actually explored previously). We regret not commenting on this point in the initial submission. In response, we have included both an analysis of the compartmentalization of the networks used in this work and a discussion on the apparent contrast between such observations to the manuscript (L215–219). Specifically, we found that the networks in our database are generally compartmentalized, which suggests that the observed backbones must exist within modules, explaining some of the noise present in our results.

5) *Number of backbones.*

figure 3: How do the 25% of the networks look like that do not contain the three backbones presented here? Is there perhaps a fourth or even a fifth backbone? How many backbones do we need to explain 100% of the empirical data and how many potential backbones with 6 links exist?

R: We thank the Reviewer for this comment, which, in combination with the comment regarding backbone size (first comment made by the Reviewer), it has been the main motivation behind the major change made to the manuscript. The main purpose of our previous figure 3 was showing that, with only few link removals, three structures could explain the backbones found for every empirical network (with two link removals one would actually explain 100% of those). In hindsight, however, we now realize that this figure (as well as the method used), was not the most straightforward way to visualize the backbones, and we acknowledge that the actual number of backbones picked was somewhat arbitrary. As mentioned in the first comment, we now use clustering techniques on the dissimilarity matrices found as a result of the alignment of the n -link backbones. In particular, regardless of the size n of these backbones, we consistently found that just two structures can capture most of the variability underlying our dataset. Notably, the number of actual backbones it is now something that comes directly out of the analysis, and our new figure 3 characterizes the variability associated with these structures.

Furthermore, the new alignments of backbones also allowed us to improve the visualization of the backbones. In our initial submission, we chose the three substructures out of every network's backbone that better represented the variation in our dataset. The problem with this approach was that variation in the shape of the backbones across networks was not represented and some of the information was lost, potentially misleading the reader. Now, the two backbones found for our dataset are generated using the backbone alignments. To do so, we follow the idea presented in the illustrative example shown in Fig. 1C of the main text. For any given size n , we selected the medoid for each of the two clusters of our dissimilarity matrix and measured the overlap with all the other "within-group" backbones. This produced two structures for any given backbone size n that best characterize the clusters, where the weight of the links represents the likelihood of finding them (L161–176). We expect that the changes made to the manuscript will clear up the Reviewer's concerns and thank them again for such useful comments!

6) *Backbones in different ecosystems.*

figure 3 / S6: I asked myself whether the three backbones that you present in the main article (1= something nested, 2= something with omnivory and 3= something broom-like) are typical representatives for different ecosystems, but could find no answer. I was looking for something like figure S6 but based on the three backbones presented in figure 3. Instead, figure S6 left the impression that your results are actually not very robust, since every ecosystem has its own three backbones instead of being represent by the ones that are introduced in the figure 3. This left me very puzzled...

R: We thank the Reviewer for his comment, which has also been one of the motivations leading to the most pronounced changes to our manuscript. We now realize that the

comparison of the backbones through figure S6 was problematic because the number of networks for each ecosystem is different. Considering that the backbones were generated as the few structures out of every network's backbone that better represented the variation in our dataset, we must expect a lot of variability when considering individual ecosystems separately. That said, the changes made to the manuscript regarding the way we compare the backbones across networks have, we believe, substantially reduced this variability. In particular, a key difference is that the structures we use to characterize the backbones are now the result of the clustering analyses on the dissimilarity matrices for each ecosystem. Therefore, these structures are now the product of overlapping all the backbones from each cluster to their respective medoids. Fortunately, this chance also provided a more appropriate way to *statistically* test whether or not the backbones are typical representatives of any ecosystem type. To do so, we compared the clustering found for the entire dataset to every separate ecosystem, and found this to be a strong predictor of the variation therein (L64–75 of the Supplementary Information).

7) *Backbones as a toy model to understand the effects of disturbances and climate change.*

*I had an additional idea for a possible application of backbone structures that might be worth mentioning in the discussion. Many researchers currently focus on the question how ecosystems react to disturbances and scenarios of current global change. There are basically three approaches to do this: (1) Simply construct a specific network based on empirical data from a specific study site and analyse its response to changing conditions. Unfortunately, the results are then only relevant for this specific network without revealing any general insights. (2) Use an ensemble of abstract networks (as for example done in Binzer, Amrei, et al. "Interactive effects of warming, eutrophication and size structure: impacts on biodiversity and food-web structure." *Global change biology* 22.1 (2016): 220-227.) in order to develop a more general understanding of the processes at hand. Unfortunately, such a study can become quite costly in terms of computer run-time. (3) Use a toy model that represents a universal, easy-to-handle food web module instead of complex food webs. The most prominent candidate is a food chain or a simple predator-prey system (as for example done in Fussmann, Katarina E., et al. "Ecological stability in response to warming." *Nature Climate Change* 4.3 (2014): 206.), which probably oversimplifies the whole problem. So using a set of network backbones might be a good compromise between being too specific, too complex or too simple. It might provide a good basic understanding of the relevant processes and thus lead to more reliable predictions.*

R: We believe that this suggestion is, in a way, related to idea of a backbone of interactions as an internal motor that is driving the dynamics of complex ecological communities (L232–239). Therefore, we included the Reviewer's suggestion to the discussion of

the revised manuscript following such perspective (L241–243) as it adds further context to the potential relevance of our results.

8) *Backbones of randomized networks.*

figure 3: I had the feeling that backbones of only six links are actually too small to be meaningful. So I was thinking about the following test: Could you also check for randomized networks in addition to randomized alignments? More precisely, if you construct “random networks” by reshuffling the interactions within the real networks, would these randomized networks still contain the same backbones? And does the result depend on the algorithm of the reshuffling (e.g. keeping the number of predators or preys or links per species constant)? I think the backbones that you found would greatly gain relevance if you could show that they only occur within real but not within random networks.

R: We thank the reviewer for this comment, and we share interest on answering such question. Unfortunately, there are substantive computational limitations to our approach that make this analysis out of reach for this study. Aligning a pair of networks can take from few minutes to many hours depending on the size of the networks being aligned; and aligning our 357 networks implies performing 63546 pairwise optimizations. Following this, randomizing those networks and aligning them again means repeating this process multiple times. The actual problem, however, becomes evident when we compare the backbones found across networks. That is, comparing the backbones found for 357 networks means performing again 63546 alignments, but comparing the real backbones and a single set of 357 randomized networks implies performing 254541 alignments. For every randomization of our dataset, the number of alignments increases quadratically. With this in mind, performing a meaningful comparison between the backbones in real and random networks becomes very challenging (at best). Despite these difficulties, we believe the Reviewer’s question is scientifically very interesting and actually see it as one of the next key steps to be addressed. For example, we think that an interesting idea would be to progressively constrain the randomizations to understand when the backbones emerge. Are there any species properties that determine the existence of the backbones? Similarly, if the backbones are maintained following a randomization—e.g. preserving the degree distribution of the species—but the properties such as the motif composition are significantly changing, it would mean that it is the periphery attached to the backbone the one absorbing the effects of the reshuffling. We included in the new version of the manuscript a brief discussion regarding this aspect (L261–267).

9) *Minor comments.*

In the first paragraph of the introduction, you list a number of prominent examples of

processes and disturbances that influence the structure of ecological networks and that potentially explain their differences. To my knowledge, the network structure might also change over time due to intrinsic processes, even in the absence of disturbances and spatial or temporal variability, leading to a broad range of possible network structures within one and the same (undisturbed) environment - See for example Allhoff, Korinna Theresa, et al. "Evolutionary food web model based on body masses gives realistic networks with permanent species turnover." Scientific reports 5 (2015).

R: We agree with the Reviewer about this comment and we have included this idea in the first paragraph of the introduction (L23–24).

Within the same model (Allhoff 2015), it has been shown that the network structure not necessarily influences its functioning. I wonder whether this could be explained in terms of a network backbone that stays intact during ongoing species turnover and that ensures the overall functioning of the web...? See Allhoff, K. T., and B. Drossel. "Biodiversity and ecosystem functioning in evolving food webs." Phil. Trans. R. Soc. B 371.1694 (2016): 20150281.

R: We thank the Reviewer for this suggestion that again relates to the idea of a backbone as an internal motor driving the dynamics of complex ecological communities. We added the Reviewer's suggestion in the same paragraph of the discussion (L239–241).

First paragraph of the introduction: Another thing that potentially influences the structure of (empirical) food webs is the sampling method. Some methods might ignore weak links, which could be captured using other techniques, leading to completely different networks when using different methods...

R: We agree with the Reviewer that this is an important aspect that needs to be mentioned in the first paragraph; therefore, we added this to the introduction of the new version of the manuscript (L29–31).

line 132: How do you define "best"? If I understood it correctly, then you use simulated annealing to get the best alignment between two networks and then you simply pick the species that is best aligned. But how do you make sure that your annealing algorithm is not stuck within a local extremum? Maybe another alignment would be even better?

R: Although it is true that in any optimization process there is always the possibility of getting stuck in a local minima, the simulated annealing algorithm is specially designed to reduce the likelihood of this happening. However, we agree with the Reviewer that multiple alignments of the same pair of networks would have been much better. The reason is because networks can have symmetries within their structure; therefore, different alignments can produce the same value for the cost function. Indeed, we performed an analysis of the alignment variability across alignments, showing that the number of

equivalent pairings per species is actually quite low (Supplementary Information). That said, although aligning multiple times every pair of networks is a compelling idea, we expect that those alignments would have only reduced the noise observed in our already significant results. Combined with the fact that aligning networks multiple times is computationally very expensive, we decided to limit ourselves to single alignments across all pair of networks.

line 206–209: It is not clear to me how simply aligning antagonistic and mutualistic networks might be of any use. It is well known that their structures differ considerably (see for example Thébault, Elisa, and Colin Fontaine. “Stability of ecological communities and the architecture of mutualistic and trophic networks.” Science 329.5993 (2010): 853-856.), suggesting that each of the mentioned network types would have a unique set of backbone structures...

R: We agree with the Reviewer that the differences between antagonistic and mutualistic network are already well known. In retrospect, we now realize that this paragraph was at risk of being misleading, potentially overstating the applications of the alignment technique presented in the manuscript. Therefore, we decided to change the wording of the paragraph (L254–257).

Concerning the material and methods section: I have to say that this section is remarkably nice and clear, although the study design is far from being intuitive for a non-specialist. Nevertheless, I would like to make two suggestions. (1) Unless I missed it, the definition of a species “role” is not given in the main text, which forces the reader to go back and forth between the main document and the supplementary material to understand the methods. Would it be possible to include the definition of the role into the main text? (2) I had to go back and forth between the results and the methods section. If possible, I would move the methods before the results to avoid too much scrolling.

R: These are very fair points. Regarding the definition of species role, this issue was picked up by both Reviewers, and we regret not picking up on it ourselves before submission. We have now included this definition in the Materials and Methods section of the main text (L285–296). Regarding the position of the Materials and Methods section in the manuscript, we must argue that some of the concepts described there are hard to understand without the right context. That is, we believe that reading the Methods after the Introduction might be confusing for the reader; that is, the Results section provides context for some of the key concepts described in the Methods section.

line 290-292: This is not perfectly clear to me: To get an isomorphic substructure, links have to be removed from both C and D, right? The sentence suggest that links can be removed from C or D...

R: We thank the Reviewer for making this point. We agree, in hindsight, that the definition was originally rather confusing. We have modified the wording for the definition of this measure (L36–38 of the Supplementary Information), and we are optimistic that the new version will be more to the Reviewer’s satisfaction.

figure 2: I was surprised to see a log-scale on the x-axis. If you simply rank all species in the network, then why would you need a log-scale? Apart from that, the figure suggests that your networks contain less than 100 species, although in line 222 you state that you have between 5 and 130 species per network. Could you please clarify this?

R: We thank the Reviewer for these comments. Regarding the log-scale on the x-axis, we decided to use such scale because in the paper we focus on the best aligned species. Considering that only some networks contain 80 species, for example, we thought that the log-scale better characterize the distribution of pairwise alignments. Regarding the number of species ranked in the x axis, as stated in the captions for the figures, every point in these figures indicates the median across at least 250 species. There are not enough networks with more than 100 species to be able to generate an extra point in the graph and, therefore, we excluded those species. We regret not having mentioned this in our initial submission, since it certainly leads to confusion. We have now changed the captions for the figures in the main text and Supplementary Information.

line 4-5 in the supplementary material: The four alignment quality measures have not yet been introduced, so the reference to “above” is confusing.

R: We would like to apologize for the wording, this is clearly an oversight. We have now changed the wording of the Supplementary Information (L5–6).

line 46 in the supplementary material: As indicated above, I am not convinced that you found “similar candidate backbones” across all ecosystems. Most of them can not be categorized into something nested, something with omnivory and something broom-like...

R: Following the changes in the manuscript described above and the new analysis added in the Supplementary Information, we anticipate that the Reviewer will now be more convinced by our results.

figure S1: Can you give an interpretation of the two axis?

R: The axes of figure S1 are representative of a distance space and therefore do not have a simple interpretation as would exist for something like a PCA.

Is figure S3 referenced somewhere in the text?

R: Although figure S3 was referenced in line 119 in the main text of the initial submission, it should have been referenced in line 118 as well. We thank the Reviewer for commenting on this! In the new version of the manuscript, this figure is referenced in lines 123 and 125.

figure S5: Do you have an explanation why stream and terrestrial ecosystems are better described by the three backbones than the other ecosystems?

R: Unfortunately, we do not really know the reason for the observed differences. However, these differences across ecosystems could just be a byproduct of the way we visualized the backbones in our initial submission. To avoid overreaching we decided not to speculate on this too strongly and have removed this figure from the manuscript.

line 68-69 in the supplementary material: In principle, you could also consider more complicated modules – why is 3 a good size?

R: We thank the Reviewer for this comment and regret not having commented on that in our first submission. It is true that one could consider more complicated modules. However, considering the motifs made of four species already implies considering 199 different patterns of interaction. In addition, given the alignment technique used in the manuscript—where species are aligned based on their neighbors motif-role profiles—the information associated with the addition of four-species motifs to the analysis is likely to be redundant and only slow down the optimization process. We have now included a clarification regarding this aspect in the Supplementary Information (L120–122).

line 96 in the supplementary material: You previously used a different symbol for NULL.

R: We thank the Reviewer for picking up on this. We have now changed the symbol to match the notation used in the main text.

line 100 in the supplementary material: “in Supplementary Fig.” is missing in the sentence. Same in line 104.

R: We have corrected this oversight in the current version of the manuscript.

Reply to Reviewer #2

The present study develops a new alignment technique specifically designed for directed networks such as food webs, which identifies “connected species within every network that play similar ecological roles and that also interact with each other in a similar manner”. The authors use this technique to both evaluate differences among 357 food webs across estuaries, lakes, marine, streams and terrestrial ecosystems and also to identify the “backbone of interactions” shared across the food webs. The developed technique is novel and it could greatly influence thinking in the field of ecological networks. Overall, I think this work could become a great contribution to our field.

Reviewer 2 recognized the ecological novelty of the technique used in the paper and thought that this could be a great contribution to the field of ecological networks. However, the Reviewer argued that a better description of some key concepts in the main text was needed and suggested some places in which the manuscript could be improved. We thank the Reviewer for their comments and we believe that a stronger manuscript has been produced because of them. We have addressed each of those comments and respond to them below (preceded by **R:**).

Please note that, following the revisions in response to both Reviewers, we have made a major change to one of the analyses presented in the manuscript. This change concerns the final analysis of the Results, where we identify and visualize the backbone of interactions. In particular, we improved the robustness of our study by altering the method with which we identify the backbones of interactions. We also incorporate a few additional analyses that help understand the nature of the observed backbones. We hope that these changes will be to the Reviewer’s satisfaction.

1) *Conceptual definition of role distance.*

However, the explanation of the technique and results needs to improve. Some key concepts are not well defined (or not defined at all) in the main text and the reader needs to go back and forth several times in between the main text, the Materials and Methods section and the Supporting Information to grasp the key concepts behind the technique and the axes of the main figures. In particular, “role distance” must be CONCEPTUALLY defined in the main text (somewhere in L63–70) and in the figure legends where it appears. It is not enough to send the reader to a mathematical definition in the Supporting Information. I would like to see the same conceptual work that the authors did for “backbone of interactions” in L44–46 applied to “role distance”. In fact, I do not think that “distance” is the right term for expressing the intuition behind the concept. I do not see how we could talk about distances between species roles. It seems to me that role similarity would make more sense (making the corresponding adjustments in the formula of Eq. 1 of the Supporting Information and in the figure axes). Besides, the

authors should include the mathematical definition of “role distance” (or similarity?) in the Material and Methods section.

R: We thank the Reviewer for making this point. First of all, we agree that the term “role similarity” is much more intuitive and a much better fit for the concept described in the manuscript. In the new version of the manuscript, we have changed this term and adjusted the equations accordingly. In addition, as requested by both Reviewers, we have included a mathematical definition of the role similarity between species in the Materials and Methods section of the main text (L285–296). In hindsight, the need for the definition of the concept in the main text is obvious, and we regret not incorporating it in our initial submission. Finally, given the importance of the definition of role similarity and its reiterated use in our manuscript, we have also conceptually defined it in the introduction of the main text (L70–72) as requested by the Reviewer.

2) Clarification of the ‘link removals’ measure.

A second concept that needs clarification is “Link removal”. The authors mention it in L149–150 and it appears on the x-axis and legend of Fig 3 with a “(see Material and Methods)”. However, the procedure of link removal does not appear in the Materials and Methods section.

R: The reviewer is exactly correct for pointing out this oversight. In retrospect, we realized that the incorporation of this new measure for the identification of the backbone is in itself confusing. Therefore, given the comments made by both Reviewers, we decided to change the way we identify the backbones (L153–176) and relegate the analysis with the measure of “link removals” to the Supplementary Information. Regarding the specific comment made by the Reviewer, however, we also made an effort to clarify the procedure of link removals there, and for their benefit reproduce it here. The underlying idea behind this measure is to estimate the number of links that differentiate two structures. Given two structures made of n links, we recursively try all the potential k -link-removal permutations on both networks until we find isomorphic structures. That is, for one link removal, for example, we would test whether or not there is any combination of one-link removals that produces two isomorphic structures made of $n - 1$ links. With this in mind, we have now changed the description of the link removals measure to clarify its procedure (L36–43 of the Supplementary Information).

3) Conceptual definition of alignment transitivity.

A third concept that needs to be clarified in the main text is “Transitivity of species’ alignments”. I suggest providing a conceptual definition in L127 and in the legend of Fig. 1. So the reader can understand the main results and conclusions without going to the Supporting Information.

R: We agree with the Reviewer on the benefit of adding a conceptual definition of transitivity of species' alignments in the main text. The concept of transitivity of species' alignment is now conceptually defined in lines 132–133 of the new version of the manuscript (as well as in the Materials and Methods section of the main text), as suggested by the Reviewer.

4) *Minor comments.*

L104–105: clarify

R: We changed the term 'role distance' to 'role similarity' as well as slightly modified the wording of the text to clarify the sentence (L109–111). In addition, the concept of role similarity is now previously defined in the introduction, which helps the reader understand the underlying idea of the sentence.

L127–130: Explain why the result “We observed that the best aligned species show a significantly higher alignment transitivity than would be expected at random (Fig. 2)” implies “the best aligned species for the different food webs are in fact paired with each other more often than expected by chance”, and why this “satisfies condition (1)”. A conceptual definition of alignment transitivity would help to clarify this result.

R: As suggested by the reviewer, we added a conceptual definition of alignment transitivity before presenting this result (L132–133). We are confident that the Reviewer will now find the writing much more convincing.

L133–136: Same problem as in my previous comment on L127–130. Clarify.

R: As in the previous comment, we added a conceptual definition of the measure of “the path likelihood between species” to clarify the writing (L138–139).

L142: Add something like “across all the webs” after “species' pairings”

R: We agree with the Reviewer that adding “across all the webs” clarifies the sentence. We implemented the Reviewer's suggestion to the new version of the paper (L150).

L149: Remove “zero or”

R: We agree with the Reviewer's suggestion and modified the wording in the new version of the paper. Notice that due to the changes made to the manuscript, the original sentence is now in the Supplementary Information (L49–50).

L160–165: I do not understand how the identified backbones of relatively few species and interactions can explain macroscopic properties such as connectance, links per species, etc. Explain.

R: We agree with the Reviewer that backbones of relatively few species might not necessarily shed light on some of the common macroscopic properties observed across networks. Indeed, the size chosen for the backbones is a concern that was raised by both Reviewers. In hindsight, we regret not further exploring the existence of bigger backbones of interactions in our initial submission. We have now included an extended analysis of the size of the backbones that we hope will be to the Reviewer's satisfaction (L153–176).

Add letter a) and b) to distinguish top and bottom panels in Fig. 2

R: We added the reviewers suggestion to the Figure.

Supplementary Figure 7b, 4th row: B and b are not Nbrs. of D and d, respectively... shouldn't it be e instead?

R: We thank the Reviewer for picking up on this error. We have changed the Supplementary Figure to correct this (Supplementary Fig. 10B)

Reviewers' comments:

Reviewer #1 (Remarks to the Author):

It was a pleasure to read this manuscript again! All my major concerns have been addressed in a convincing way so the manuscript is now, in my opinion, highly interesting, inspiring, and definitely worth publishing. My remaining remarks are all minor.

- Looking at Fig 3, S7, S9, I find it very interesting that all these backbone structures can be roughly described as "something with high centrality: few species in the center are eaten by many satellite species" (red) and "something bipartite: half of the species are consumers, the other half is eaten" (blue). Would be nice to highlight this consistency somewhere in the text, e.g. following line 176.

- Are some species in the red networks also prone of being found in the blue networks? If yes, could this be interpreted as red = resources + first trophic level and blue = first + second trophic level?

- There are now two key messages in the manuscript: (1) There is a fingerprint of the ecosystem type on the network structure. (ii) Nevertheless, all networks share a "common backbone".

I think both messages are equally important, but key message (1) is somehow hidden in the supplementary material. Is there a way of including a figure corroborating key message (1) in the main article? Maybe a subset of figures S1 and S2?

- Fig. 3: Why do you choose networks of size 6 and 8? 6 and 8 are quite close to each other. In my opinion, choosing 6 and 16 (for example) would be more impressive and corroborate the statement that the overall backbone structure is independent of its size.

- line 122: this came as a surprise, because it goes against my intuition. I would have expected that lower level species are easier to rank. All food webs have a first trophic layer, but not every network contains a fifth trophic layer...

However, both your statement and my intuition go against Fig. 3b), which suggest that basal species are the hardest to rank...

Please clarify.

- caption of Fig. 1: "In contrast, the alignment transitivity for species d is 0 because the species in the blue and green networks to which d is paired are not themselves paired." There is no species in the green network to which d is paired!

Reply to Reviewer #1

It was a pleasure to read this manuscript again! All my major concerns have been addressed in a convincing way so the manuscript is now, in my opinion, highly interesting, inspiring, and definitely worth publishing. My remaining remarks are all minor.

Reviewer #1 was pleased with the changes made to the manuscript and provided minor comments to be addressed. We summarize each of Reviewer's comments in turn below, and we hope that our responses (preceded by **R:**) are to the Reviewer's satisfaction.

Please note that we incorporated 54 network to the analysis that had been previously withdrawn from the study. Due to a correction by one of the sources of the data used in our manuscript, we had to incorporate those and rerun the analyses. Despite the additions to the analysis, the results and conclusions have not significantly changed. The addition of those networks, however, have provided us with deeper resolution for the analysis of large backbones, which allowed us to further discuss this aspect in the manuscript (L173–175) and supplementary material (L53–L59).

1) *Minor comments.*

Looking at Fig 3, S7, S9, I find it very interesting that all these backbone structures can be roughly described as “something with high centrality: few species in the center are eaten by many satellite species” (red) and “something bipartite: half of the species are consumers, the other half is eaten” (blue). Would be nice to highlight this consistency somewhere in the text, e.g. following line 176.

R: We agree with the broad description of the backbones made by the Reviewer, and we added such description to the discussion section of the manuscript (L209–212). We are optimistic that the Reviewer will find the writing convincing and thank them for the insightful comment.

Are some species in the red networks also prone of being found in the blue networks? If yes, could this be interpreted as red = resources + first trophic level and blue = first + second trophic level?

R: Following the Reviewer's comment, we studied the trophic level of the species forming each backbone. In particular, we classified species as basal (0), intermediate (1) or top (2), and we calculated the mean trophic level of the backbones across all networks. We found no significant differences in the trophic level of the two backbones. In the new version of the manuscript, we incorporated this new result to the main text (L212–214) and supplementary material (L94–99). We thank the reviewer for this interesting suggestion.

There are now two key messages in the manuscript: (1) There is a fingerprint of the ecosystem type on the network structure. (ii) Nevertheless, all networks share a “common backbone”. I think both messages are equally important, but key message (1) is somehow hidden in the supplementary material. Is there a way of including a figure corroborating key message (1) in the main article? Maybe a subset of figures S1 and S2?

R: In hindsight, we acknowledge that the first key message might have ended up somewhat hidden in the supplementary material, and we agree with the Reviewer that adding a figure to the main text would be helpful. Following this, we added supplementary figure 1 to the main text.

Fig. 3: Why do you choose networks of size 6 and 8? 6 and 8 are quite close to each other. In my opinion, choosing 6 and 16 (for example) would be more impressive and corroborate the statement that the overall backbone structure is independent of its size.

R: We agree with the Reviewer that choosing backbones of size 6 and 16 could be more impressive. Note that, after adding the 54 previously withdrawn networks to the analysis, we have now extended the discussion regarding the size k of the backbones. In particular, we found that the differences between the observed backbones are much less evident when $k > 15$. Importantly, we believe that this threshold could represent a size limit for the identification of the backbones in our dataset. Following this, we changed Fig. 3 of the main text to include the backbones of size 6 and 15, and we added a discussion of the size limitations for the backbones in the main text (L173–175) and supplementary material (L53–L59).

line 122: this came as a surprise, because it goes against my intuition. I would have expected that lower level species are easier to rank. All food webs have a first trophic layer, but not every network contains a fifth trophic layer... However, both your statement and my intuition go against Fig. 3b), which suggest that basal species are the hardest to rank... Please clarify.

R: We thank the Reviewer for this comment and apologize for the lack of clarity regarding this aspect of the analysis. First of all, we broadly classify species as ‘basal’ (i.e. it does not feed on any species in the network), ‘top’ (i.e. it is not consumed by any species in the network), and ‘intermediate’ (e.i. it feeds on and is consumed by other species in the web). Therefore, we do not distinguish between five trophic layers, with all the networks containing all three levels described. In addition, given such classification, intermediate species are indeed the most abundant. Following this, we changed the wording of the caption for supplementary figure 3B to avoid any confusion. Perhaps more importantly, the fact that a trophic level is overrepresented relative to others

does not imply that it is going to be easier to align (or rank). Imagine, for example, a dataset where all food webs contain a single top predator. When aligning those networks, top predators would likely be the easiest to align and rank because there would not be other species playing the same role in their respective networks. That said, we slightly changed the wording of the results section to clarify this point (L122). We hope that these changes will be to the Reviewer's satisfaction.

Caption of Fig. 1: "In contrast, the alignment transitivity for species d is 0 because the species in the blue and green networks to which d is paired are not themselves paired." There is no species in the green network to which d is paired!

R: We thank the Reviewer for picking up on this, we have corrected the caption of this figure in the next version of the manuscript.

REVIEWERS' COMMENTS:

Reviewer #1 (Remarks to the Author):

I was very pleased to read this manuscript again. As stated in the last two review rounds, I am very excited about the approach and the results presented in this study. All my major concerns of the last review round have been addressed convincingly.

Nevertheless, after scanning through the document once again, I still have one more question to ask. The authors often refer to 6-link-backbones, but the pictures shown in the document have actually more than 6 links (see for example Figure 4 A, but the same problem also occurs in Figure 4B with $k=15$ and in several other figures in the appendix). In fact, not even the two backbones presented for one value of k have the same number of links, indicating that this is more than a typo... Please clarify!

Reply to Reviewer #1

I was very pleased to read this manuscript again. As stated in the last two review rounds, I am very excited about the approach and the results presented in this study. All my major concerns of the last review round have been addressed convincingly.

Reviewer #1 was pleased with the changes made to the manuscript and provided a minor comment to be addressed. We answer the Reviewer's comment below, and we hope that our response (preceded by **R:**) is to the Reviewer's satisfaction.

1) *Minor comments.*

Nevertheless, after scanning through the document once again, I still have one more question to ask. The authors often refer to 6-link-backbones, but the pictures shown in the document have actually more than 6 links (see for example Figure 4 A, but the same problem also occurs in Figure 4B with $k = 15$ and in several other figures in the appendix). In fact, not even the two backbones presented for one value of k have the same number of links, indicating that this is more than a typo... Please clarify!

R: In contrast to figure S7, in figures 4, S6 and S9, we do not represent examples of k -link backbones but overall descriptions of the two *representative* k -link backbones found across our dataset. As stated in the caption of figure 4, given the backbones found for every network of our dataset, the representative k -link structures are found by overlapping all the within-cluster backbones as shown in Fig. 1C of the main text. Therefore, the links of the networks depicted in figures 4, S6 and S9 represent a summary of the interactions forming the k -link backbones across networks, where the weight of the links is proportional to the likelihood of finding them in the different backbones. In order to avoid this or related confusion going forward, we have rephrased the captions of figures 4, S6 and S9 to clarify this aspect. We thank the Reviewer for raising this point.